# Evaluation of Three-Dimensional Bioprinted Human Cartilage Powder Combined with Micronized Subcutaneous Adipose Tissues for the Repair of Osteochondral Defects in Beagle Dogs

**DOI:** 10.3390/ijms23052743

**Published:** 2022-03-01

**Authors:** Jina Ryu, Mats Brittberg, Bomi Nam, Jinyeong Chae, Minju Kim, Yhan Colon Iban, Martin Magneli, Eiji Takahashi, Bharti Khurana, Charles R. Bragdon

**Affiliations:** 1ROKIT Healthcare Inc., Seoul 08514, Korea; jina.ryu@rokit.co.kr (J.R.); bomi.nam@rokit.co.kr (B.N.); jinyeong.chae@rokit.co.kr (J.C.); minju.kim@rokit.co.kr (M.K.); 2Cartilage Research Unit, Joint Restoration Center, University of Gothenburg, 41345 Gothenburg, Sweden; mats.brittberg@regionhalland.se; 3Region Halland Orthopaedics, Kungsbaka Hospital, 43440 Kungsbacka, Sweden; 4Harris Orthopaedic Laboratory, Massachusetts General Hospital, Boston, MA 02114, USA; colonibany@gmail.com (Y.C.I.); martin.j.magneli@gmail.com (M.M.); eiji@kanazawa-med.ac.jp (E.T.); 5Department of Orthopaedic Surgery, Harvard Medical School, Boston, MA 02115, USA; 6Department of Clinical Sciences, Danderyd Hospital, Karolinska Institute, 18288 Stockholm, Sweden; 7Department of Orthopaedic Surgery, Kanazawa Medical University, Ishikawa 920-0293, Japan; 8Department of Radiology, Brigham and Women’s Hospital, Harvard Medical School, Boston, MA 02115, USA; bkhurana@bwh.harvard.edu

**Keywords:** bioink, cartilage, extracellular matrix, osteoarthritis, 3D printing

## Abstract

Cartilage lesions are difficult to repair due to low vascular distribution and may progress into osteoarthritis. Despite numerous attempts in the past, there is no proven method to regenerate hyaline cartilage. The purpose of this study was to investigate the ability to use a 3D printed biomatrix to repair a critical size femoral chondral defect using a canine weight-bearing model. The biomatrix was comprised of human costal-derived cartilage powder, micronized adipose tissue, and fibrin glue. Bilateral femoral condyle defects were treated on 12 mature beagles staged 12 weeks apart. Four groups, one control and three experimental, were used. Animals were euthanized at 32 weeks to collect samples. Significant differences between control and experimental groups were found in both regeneration pattern and tissue composition. In results, we observed that the experimental group with the treatment with cartilage powder and adipose tissue alleviated the inflammatory response. Moreover, it was found that the MOCART score was higher, and cartilage repair was more organized than in the other groups, suggesting that a combination of cartilage powder and adipose tissue has the potential to repair cartilage with a similarity to normal cartilage. Microscopically, there was a well-defined cartilage-like structure in which the mid junction below the surface layer was surrounded by a matrix composed of collagen type I, II, and proteoglycans. MRI examination revealed significant reduction of the inflammation level and progression of a cartilage-like growth in the experimental group. This canine study suggests a promising new surgical treatment for cartilage lesions.

## 1. Introduction

According to the prevalence report published in 2017 from the US Center for Disease Control and Prevention (CDC), approximately 23% of US adults were diagnosed with arthritis during 2013–2015 [1]. This represents 54.4 million citizens, of which 44% are reported to have limitations in daily activities due to the disease. The incidence of osteoarthritis (OA) in the US is expected to increase to 78 million by 2040 [2]. Patients with OA have a high burden of medical expenses due to mid- to long-term treatment and rehabilitation periods [3]. In 2013, OA was responsible for $164 billion US dollars in lost wages which is equivalent to a loss of $4040 per year per individual [4]. These statistics show that the burden of medical expenses increases over time, and the number of citizens experiencing loss of function and wages is increasing. This all highlights the need for the development of better treatments, from prevention, mitigation, and surgical intervention.

As with all attempts to regenerate tissue in the human body, there are three elements that are considered crucial. The first is the source of cells. Stem cells are an attractive source as they have the capacity to become any number of cell types under the right conditions [5,6]. The second is to provide a scaffold, a structure that is conducive to cell growth and differentiation that can guide the development and maturation of the tissue. Preferably, this scaffold would resorb over time, leaving behind healthy new tissue [7,8]. The third is to provide the micro-environment that is conducive to pushing the stem cells to develop into the cell type that is desired. This is the most complicated aspect of tissue engineering and incudes cell nutrition, providing the right building blocks for development and an understanding of the complex biochemical signaling that affects so many aspects of cell development [9,10].

Three-dimensional printing makes it possible to fabricate complex forms with high precision through a layer-by-layer addition of different biomaterials such as extracellular matrix (ECM) and biocompatible bioinks (e.g., gelatin methacryloyl (GelMA), methacrylated hyaluronic acid (HAMA), polycaprolac-tone (PCL), and etc.). With the use of a 3D printer, it might be possible to enhance the cell viability and to provide more tissue-like structures. In the early stage, 3D bioprinting was used for simple medical purposes such as dental implants and prosthetics. With recent advances in 3D printing technologies, the production of blood vessels and artificial oculus has recently been reported [11,12]. The main objective of 3D manufacturing is the restoration of damaged tissues or organs, and it is important to create a native complex environment (niche) for cell differentiation and tissue regeneration [13].

The purpose of this study was to investigate the ability to use a 3D printed matrix comprised of lyophilized costal-derived cartilage matrix (LCCM) and a Minimally manipulated Adipose tissue-Extracellular Matrix (MA-ECM) to repair a critical size femoral chondral defect using a canine weight-bearing model.

## 2. Results

### 2.1. Generation of Osteochondral Defects and Matrix Implantation

There were 12 mature male beagle dogs used in this study. Animals were divided into 4 groups with 3 dogs in each group:

Control Group:A defect control without any treatment,

Experimental groups:2.LCCM (lyophilized costal-derived cartilage)3.MA-ECM (Minimally manipulated Adipose tissue-Extracellular Matrix (MA-ECM)4.LCCM/MA-ECM.

To evaluate the repair at both 20 and 32 weeks, we made a defect first on the left limb, followed by the right limb after another 12 weeks (Figure 1A). In the experimental groups, the different biomaterial inks were printed by a 3D bioprinter (Figure 1B). The biomaterial ink was mixed with thrombin solution, and the resulting hardened matrix was removed from the mold and implanted on the defect of canine knee joints (Figure 1C).

### 2.2. Gross Appearance of Chondral Defect Area in Each Group

Chondral defects were observed at weeks 20 and 32 after implantation (Figure 2 and Appendix A). At the time of tissue harvest, the defects in the negative control group showed a repair tissue with fibrous, translucent tissue that was soft in texture. Within this tissue at week 20 were distinct circular zones that were less translucent and cloudy in appearance. At week 32, it appears that these zones have coalesced and developed into a whiteish soft tissue, partially filling the defects. The defects in LCCM group had a similar appearance both at week 20 and 32 to that of the control group at week 32, demonstrating a more rapid development of the fibrous tissue within this group. The defects in MA-ECM group were filled with dense whitish tissue. There were areas of apparent incorporation of the de novo tissues and the existing surrounding normal chondral tissue. The defects in LCCM/MA-ECM group were filled with tissue showing a distinctly different appearance and texture, similar to normal articular cartilage, than the tissues in the other three groups (Figure 2B). The tissue appeared more organized, having a thicker, denser, white shiny appearance. The tissue in those defects appeared to be developing centrally out to the periphery as well as appearing to be healing also from the periphery inward. In addition, there appeared to be lamellar structures within this tissue, blending with the surrounding existing normal cartilage. The repair tissue in the LCCM/MA-ECM group began developing a cartilage-like appearance at week 20. It was more similar to normal cartilage at week 32 than the other groups.

### 2.3. Serum Level of Matrix Metalloproteinase-3 and -9 (MMP-3 and -9)

Serum MMP-3 and -9 levels were analyzed to observe the degree of eventual inflammatory response after implantation [14,15,16]. As shown in Figure 3A, the serum MMP-3 levels were not significantly different until week 2. However, on week 4, although the levels in defects and LCCM were slightly increased, in the MA-ECM and LCCM/MA-ECM group, levels were significantly decreased (* *p* < 0.05).

The serum MMP-9 levels in MA-ECM and LCCM/MA-ECM groups were significantly downregulated compared to the other groups from day 1 to week 4 (Figure 3B). Interestingly, there was no significant difference between MA-ECM and LCCM/MA-ECM groups. Therefore, it can be said that the experimental groups containing MA-ECM showed lower serum MMP-3 and 9 levels, indicating its potential to reduce the inflammatory response.

### 2.4. Mechanical Testing

To test the mechanical properties of the repair cartilage, compressive strength test was conducted (Appendix A). The three controls had much softer tissues, and the values under the loading conditions for cartilage were highly variable. In those cases, the indenter pushed through the thickness of the ingrown tissue and, in contrast to the normal cartilage and LCCM/MA-ECM group, did not leave a permanent indentation upon removal. The average stiffness and average stress at the relaxation of the normal cartilage were 15.1 ± 6.8 N/mm^2^, and 0.1 ± 0.06 N. These values were not statistically different from that of LCCM/MA-ECM at either 20 or 32 weeks, 12.8 ± 3.8 N/mm^2^, 0.28 ± 0.16 N and 23.2 ± 17.9 N/mm^2^, 0.36 ± 0.20 N respectively.

### 2.5. Magnetic Resonance Imaging (MRI) Analysis

MRI scanning was performed at weeks 2, 4, 12, 20, and 32 after surgery (Figure 4A). MRI images were scored based on MOCART score [17,18]. This scoring system is commonly used to assess the improvement of the repair tissue after surgical treatment followed by the previously published classification system. Eight criteria were used to describe the morphology and signal intensity of the repair tissue compared to the adjacent native cartilage (Appendix A). The bar graph shows the total MOCART score over time for the four groups (Figure 4B). There was a strong trend towards higher MOCART scores in the LCCM/MA-ECM group compared to the defect control group, with the difference increasing over time. At week 20, the articular cartilage volume was less than 25% in the defect control group, and the filling consisted of homogeneous tissue, and the signal intensity of repair tissue was abnormal. In contrast, volume fill for LCCM/MA-ECM was relatively higher than the other groups. The filling consisted of inhomogeneous tissue. Furthermore, the signal intensity of repair tissues was slightly abnormal. After week 32, cartilage-like tissues with the same thickness as the surrounding native tissues were mostly developed in LCCM/MA-ECM group. Examples of the MRI at week 32 for the control and LCCM/MA-ECM group are shown in Figure 4A. While there was a persistence of subchondral edema in the control group, there was a more cartilage-like appearance of the tissues formed in the defects when LCCM and stem MA-ECM were used.

### 2.6. Microscopic Findings

The histological observations were assessed according to the International Cartilage Repair Society (ICRS) visual histological assessment scale for each group [19,20]. The six criteria were used to describe the morphology of the repaired cartilage (Appendix A). At week 20, the H&E staining in the control defect group indicated that the articular surface was discontinuous, consisting of fibrocartilage, and the surface cells of cartilage were clustered (Figure 5A). In contrast, the articular cartilage surface of MA-ECM and LCCM/MA-ECM groups was relatively smooth, and the matrix consisted of hyaline and fibrocartilage mixture at week 20. The defect control group showed disorganized cell distribution scoring 0 points on ICRS histological assessment scale at week 32 (Table 1) [19,20]. In contrast, the matrix of the experimental groups was observed to be hyaline-like, as indicated by cells organized in columns at week 32. While there were no significant differences among the scores of the LCCM, and MA-ECM groups, only LCCM/MA-ECM group showed significance compared to the defect control group both at week 20 and 32 (*p* < 0.05, *p* < 0.01, respectively). Interestingly, at week 32, LCCM/MA-ECM showed significantly higher scores than LCCM group (Appendix A). Although LCCM/MA-ECM implantation significantly improved the repair as seen in the visual histological scores compared to the other groups at weeks 20 and 32 after matrix implantation, it will take more than 32 weeks to regenerate into thick and healthy tissues (Figure 5B).

### 2.7. Safranin-O Staining and Immunohistochemistry (IHC)

Safranin-O staining and IHC were performed to confirm the characteristics of the newly formed tissue. After staining, the surface area in the histological section was analyzed. From the results at weeks 20 and 32, it was confirmed that the newly formed tissue in LCCM/MA-ECM group stained strongly with safranin-O compared to other groups (Figure 6A). This indicates that the cartilage-like tissue of the LCCM/MA-ECM group contained proteoglycans with similarity to normal cartilage. Type 1 collagen was observed in LCCM, MA-ECM, and LCCM/MA-ECM groups at week 20, but no type 1 collagen was found in LCCM/MA-ECM groups at week 32 (Figure 6B). As a result of IHC of type 2 collagen, type 2 collagen was detected in MA-ECM and LCCM/MA-ECM groups at week 32, while no staining was observed at both 20 and 32 in the defect group (Figure 6C). LCCM/MA-ECM implantation induced more hyaline-like cartilage compared to the other groups that showed fibrocartilaginous or fibrous tissue repairs. At week 20, the cells of the repair tissue were clustered from the bottom of the subchondral bone, and at week 32, their morphology was changed into a smooth layer shape to form a cartilage-like matrix. It was suggested that the supplementation of extracellular matrix-induced type 2 collagen-rich hyaline-like cartilage tissues, in all groups compared to the control group.

## 3. Discussion

In this study, we investigated the efficacy of a grafted biomaterial on the repair of osteochondral lesions in knee articular cartilage tissues using a canine weight-bearing model. The results of this study represent a new method of how to induce an articular cartilage repair in an animal model. One of the experimental groups showed a repair with hyaline characteristics. If those results can be replicated clinically, it would represent a viable new treatment for injured cartilage using off-the-shelf products. In order to apply the clinic, long-term viability verification of the new tissue must be demonstrated.

In contrast to the control group and the groups that contained only LCCM or MA-ECM, the combined LCCM/MA-ECM group had striking visual, histological, immunochemistry, and mechanical similarities to that of normal articular cartilage. The tissue in those defects appeared to be developing centrally out to the periphery as well as to be healing from the periphery inward. At both 20 and 32 weeks, much of the defects appeared to be occupied by new thick white cartilage-like tissue. In addition, there appeared to be lamellar structures within this tissue that blend with the existing normal cartilage. The mechanical stiffness of the new tissues in the experimental group was similar to that of normal cartilage. In contrast, the tissues of the control group and the two single compound groups were much softer and pliable at both 20 and 32 weeks, suggesting that the transformation to cartilage-like tissue was arrested or significantly delayed. Upon microscopic examination of the LCCM/MA-ECM group, there was a well-defined surface layer covering the middle layer of chondrocytes, which were surrounded by a matrix composed of type 1 and 2 collagen, and proteoglycans, all similar in appearance to normal cartilage. In addition, analysis of MRI scans of the defect area of time indicated the progression of cartilage-like tissue growth into the defects in the experimental groups.

The time-point MRI anlaysis of the defect area indicated a progression of cartilage-like tissue growth into the defects in LCCM/MA-ECM group. On the other hand, MOCART scoring results demonstrated that there was no significant difference between MA-ECM and LCCM/MA-ECM groups but the difference to other control and the LCCM group. In addition, similar patterns were observed in the serum MMP-9, a representative serum biomarker for osteoarthritis [14]. Together with these findings, a higher MOCART scoring and a lower inflammation plus a higher similarity with the normal tissues, the combination of LCCM and MA-ECM might be interesting biological sources of inducing a hyaline-like cartilage repair.

In this study, we applied a 3D printing technology to a local cartilage repair animal model. There are several previous studies using 3D printers. Beketov et al. applied bioprinting on de novo cartilage formation [21]. They printed a bioink mixed with chondrocytes and 4% collagen and conducted a small animal experiment. As a result, cartilage tissue was formed within 5–6 weeks, and it contained a high content of glycosaminoglycan and type 2 collagen. However, the use of heterogeneous collagen can induce an immune response, and when cultured chondrocytes are transplanted into the body, there are concerns about safety due to several side effects. Mancini et al. [22] printed a top layer with articular cartilage progenitor cells, ACPC, and a hydrogel, and the bottom layer was PCL and MSC. After 6 months, although significant bone growth was observed, the formed cartilage-like tissues were limited, and there was no significant difference in histological scores. Several factors, including tissue loss after transplantation and high decomposition rates of hydrogels, could influence the results. In another study, induced pluripotent stem cells (iPSCs) and nanofibrous cellulose were used [23]. After five weeks, hyaline-like cartilaginous tissue was seen with collagen type II expression, and the tissue lacked tumorigenic Oct4 expression. However, since iPSC shows unstable differentiation, more studies are still required to be used in the clinic. In this current study, cells are not transplanted directly. By transplanting the cell environment (niche) with the shape and size of the defect, intrinsic cells can be attracted to migrate to the defect to start a tissue repair.

There are several limitations to this study. The number of animals used in each group was not sufficient for more-robust statistical analysis of the collected data. For the mechanical texting, only one type of loading profile was used and the one used was designed for the analysis of an intact cartilage matrix. A different loading profile would have been needed to properly characterize the softer tissue present in the defects of the control and the LCCM, and MA-ECM groups. In those groups, the probe often penetrated the soft tissue layer to the harder sub-condral layer to varying extents, leaving no indentation mark. This is in contrast with the indentation mark left after mechanical testing in the intact cartilage and LCCM/MA-ECM tissue. There were some limits in the resolution of the MRI images. While clinical MRI images are of higher quality, the differences in the MRI grading score in this study were still striking. Histological and radiographic grading systems are all subjective measures. Finally, though we attempted to create a purely cartilaginous defect, it was clear that the subcondral bone was violated to some extent in most animals, and this can be seen in the form of microfracture repair deep to the implantation site in some sections. Although our study bears several limitations, such as the insufficient number of experimental animals and molecular markers, the overall data presented in this study at 20 and 32 weeks indicates a potential ability to induce a hyaline-like cartilage tissue under weight-bearing conditions using a 3D printed biomatrix composed of LCCM/MA-ECM.

## 4. Materials and Methods

### 4.1. Animals and Study Design

In this study, 12 mature male beagle dogs, weighing ~10 kg were used. Maturity was confirmed by radiological examination of the closure of the humoral growth plate. All experimental procedures were performed following the guidelines for the management and use of laboratory animals. All animals were euthanized at week 32 post-initial surgery giving a follow-up of 32 weeks for the left knee and 20 weeks for the right knee. The detailed study design is summarized in Table 2.

### 4.2. Preparation of Experimental Surgery

The animals followed NPO for 12 h before experimental surgeries to prevent adverse effects from general anesthesia. 30 min before each operation, they received 25 mg/kg of cefazolin IM as prophylactic antibiotics The dogs were anesthetized with 0.25 mg/kg IM of acepromazine and 4.4 mg/kg IM of Telazol (Zoetis, Troy Hills, NJ, USA) and maintained with 1–3% isoflurane as inhalation anesthesia. Then, the surgical site was shaved and disinfected with isopropyl alcohol and Chloraprep™ stick (Becton, Dickinson and Company, Franklin Lakes, NJ, USA).

### 4.3. Surgery Procedure

Surgeries were performed first on the left limb, followed 12 weeks later by the right limb. With such staged procedures, the follow-up of the repair was evaluated at both weeks 20 and 32. After sterilization and draping of the operation area, an incision of 3 cm length was conducted on the medial parapatellar region. The patella was retracted laterally through the retinaculum and articular capsule. The medial meniscus could carefully be retracted medially, which allowed a good exposure of the weight-bearing area of the medial femoral condyle.

The cartilage defect was created using a bone punch osteotome (6 mm diameter). After each mallet blow, the osteotome was rotated 90 degrees. The remaining cartilage was scraped off. The thickness of the cartilage was thin (approximately 2 mm) in all cases. Fibrin glue was placed into the defect before insertion of the 3D biomatrix. Additional fibrin glue was placed on top of the matrix to hold it in place. The medial meniscus was reduced back into anatomical position. The capsule and retinaculum were closed with resorbable sutures, and the skin incision was closed with non-absorbable sutures. After the procedure, the leg was wrapped with an elastic wrap. The 6 mm in diameter and 2 mm deep chondral defect, created in the weightbearing area of the medial femoral condyle, is shown in Figure 1.

### 4.4. Preparation of Biomaterial Ink

#### 4.4.1. MA-ECM (Minimally Manipulated Adipose Tissue-Extracellular Matrix)

Human abdominal adipose tissue was obtained from female patients ≤ 25 years of age at Massachusetts General Hospital. 30 mL of liposuction adipose tissue was distributed into three 10 mL syringes. Adipose tissue was minced, and the fiber was filtered out by using a micronizer (BSL Rest, Seoul, Korea). The cleaned tissue was washed 3 times with sterile saline into the LipoCell liposuction kit (Tiss’You S.R.L., San Marino, Italy) to drain the oil, blood and sterile saline into the waste pack. Adipose tissues from which all impurities had been removed and MA-ECM was harvested.

#### 4.4.2. Mixture of LCCM (Lyophilized Costal Cartilage Matrix) and MA-ECM

The human costal cartilage tissues used in this study were purchased from Life-Link Foundation, FL, USA, as lyophilized powder products (LCCM). 100 mg of LCCM was mixed with 1cc of MA-ECM by using a mixing tube.

### 4.5. Three-Dimensional Printing Procedure

#### 4.5.1. Fabrication of PCL Supports

A polycaprolactone (PCL) pellet was dissolved in a printable liquid state at 80 °C. The PCL supports were used for physical support while the printed patch was cured. PCL was constructed by extruding through a 0.4 mm diameter nozzle and printed to create a 6 mm diameter internal dimension to match the size of the defect created during surgery.

#### 4.5.2. Biomatrix Printing

For 3D printing, Dr. INVIVO (ROKIT Healthcare, Seoul, Republic of Korea, Figure 6C) was used. To solidify the biomaterial inks, Beriplast^®^ fibrin glue (CSL Behring GmbH, Marburg, Germany) was used. Biomaterial ink was mixed with 0.5 cc of fibrinogen solution by use of a mixing tube. Then, thrombin solution was printed sequentially after the biomaterial printing. After the biomatrix was injected into the printed mold, it was allowed to cure for about 5 min. The 3D biomatrix was transported to the operating room in sterile saline and was implanted directly into the defect area using additional fibrin glue to hold it in place.

### 4.6. Magnetic Resonance Imaging (MRI) Examination and MOCART Analysis

To monitor the repair of osteochondral defect region over time, MRI scanning was conducted post-operatively at weeks 2, 4, 12, 20, and 32. All MRI exams were analyzed by a musculoskeletal fellowship-trained radiologist with 16 years of clinical experience. The radiologist was blinded to the study group identification. A modified MOCART (Magnetic Resonance Observation of Cartilage Repair Tissue) 2.0 knee score consisting of seven variables

Volume fill of cartilage defectIntegration into adjacent cartilageSurface of the repair tissueStructure of repair tissueSignal intensity of the repair tissueBony defect or bony overgrowthSubchondral changes, ranging from 0 to 100 points was used to evaluate the cartilage repair site [17,18]. Additionally, the presence or absence of joint effusion was also recorded for each knee exam.

### 4.7. Measurement of Compressive Strength

Compressive strength test was conducted on all samples immediately after euthanasia, following the method described previously [24]. Using an MTS tabletop load frame (MTS, Eden Prairie, MN, USA), samples were loaded under force control at a rate of 0.1 mm/sec to a peak load of 3 Newtons. Relaxation at peak load was recorded for 900 s. Data were collected at 20 hertz. Stiffness was calculated from the slope of the loading portion of the data and the stress at relaxation was determined using the last 10 s of the relaxation data.

### 4.8. *Histology and Immunohistochemistry (IHC) Analysis*

After compressive strength test, all samples were rinsed with phosphate-buffered saline (PBS) and they were fixed in 10% neutral buffered formalin at 25–30 °C. After decalcification in 15% EDTA (ethylenediamine-tetra acetic acid) (pH 7.2–7.4), samples were dehydrated with multi-series of ethanol and embedded in paraffin and sectioned into 5-μm slides. Samples were stained with hematoxylin and Eosin (H&E), Safranin-O for proteoglycans. The histological H&E staining slides were scored with ICRS visual histological assessment scale. For immunohistochemistry, the paraffin-embedded samples were dewaxed by xylene. After blocking and antigen retrieval, all sections for immunohistochemistry staining were incubated with Rabbit type Collagen I antibody (1:100, orb213757, Biorbyt, Cambridge, UK) and Anti-Collagen II antibody (1:200, ab34712, Abcam, Cambridge, MA, USA) overnight at 4 °C. The sections were rinsed with PBS for 3 times. Finally, they were incubated with horseradish peroxidase-conjugated IgG and stained using a reagent of 3,3–20 diaminobenzidine solution containing 0.01% hydrogen peroxide. The counterstaining was completed with hematoxylin. The safranin-O staining and immunohistochemistry slides were assessed with scoring system. ICRS scale assessment criteria were used, and the highest score (3) was applied to the ideal repair result, whereas the lowest score (0) was applied to the poorest repair result. The 6 criteria are listed as below [19,20].

I.Surface. A smooth, continuous surface is an essential feature of the normal joint.II.Matrix. The unique combination of collagen and proteoglycans in hyaline cartilage provides the correct viscoelastic properties for the articular surface.III.Cell distribution. A columnar distribution of cells in the middle and lower zone of the cartilage layer indicates normal maturation, whereas disruption of this alignment indicates abnormal maturation.IV.Cell population viability. A viable cell population is essential for matrix turnover.V.Subchondral bone. The subchondral bone determines the geometry of the joint and therefore the pattern of its loading.VI.Mineralization. Mineralization within the cartilage layer is a pathological phenomenon and is indicative of functional impairment.

### 4.9. Enzyme-Linked Immunosorbent Assay (ELISA) Analysis of Metalloproteinase-3 and -9

A series of post-operative blood draws were conducted to measure the degree of inflammation at days 1, 3, and weeks 1, 2, 4. Serum metalloproteinase-3 and -9 (MMP-3 and -9) were measured to estimate systemic inflammation level. Blood samples were collected from all subjects in control and experimental groups. Using commercial MMP-3 (MBS043123) and -9 (MBS736101) ELISA kit purchased from Mybiosource (San Diego, CA, USA), we detected absorbance at 450 nm of MMPs using a microplate reader following the manufacturer’s instructions.

## 5. Conclusions

This study produced evidence of a hyaline-like osteochondral repair in a canine model using a 3D printed biocomposite. Long-term canine preclinical studies and well controlled clinical trials are needed to further evaluate this method as a clinical treatment.

## Figures and Tables

**Figure 1 ijms-23-02743-f001:**
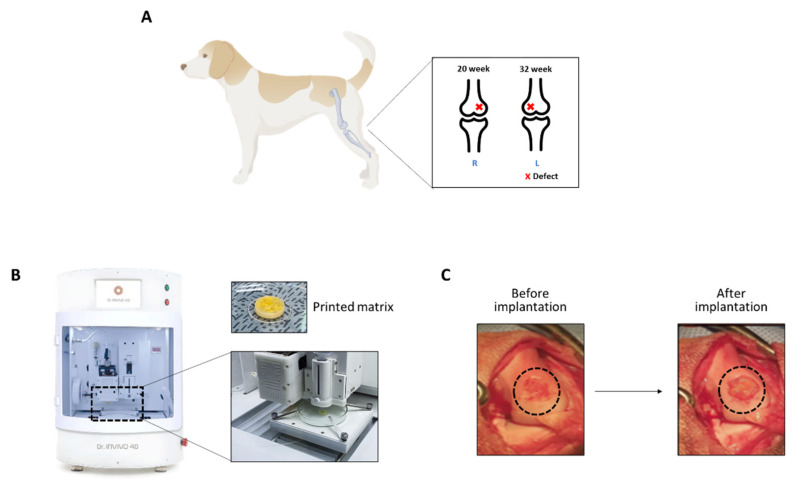
Generation of osteochondral defects in canine knee joints. (**A**) The illustration of canine cartilage defect models. Cartilage defects were created using a 6 mm bone punch on medial femoral condyles bilaterally. (**B**) 3D bioprinter (Dr. INVIVO) used in this study to print the matrix. (**C**) The printed matrix was implanted on the defect.

**Figure 2 ijms-23-02743-f002:**
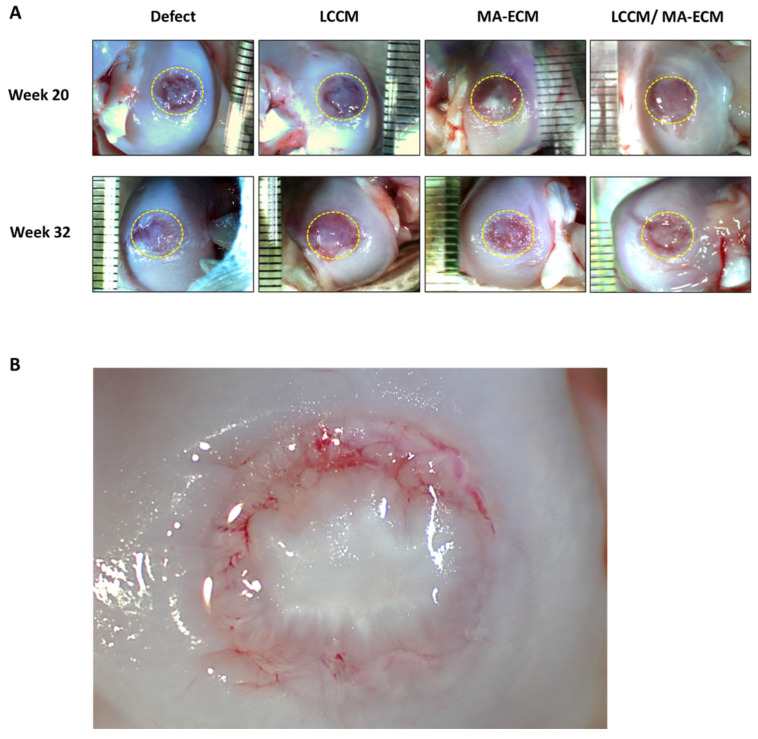
Gross appearance of the femoral cartilages. (**A**) The appearance of the cartilages after bilateral 3D-printed matrix implantation procedure performed on 12 male beagle dogs. The circle indicates a defect area with a diameter of 6 mm in each group. (**B**) The appearance of the ingrowth tissue at 20 weeks in LCCM/MA-ECM group was distinctly different than the control group, having a glistening white solid appearance similar to the surrounding normal cartilage. Lamellar structures can be seen in the new tissue, and there is good incorporation of the tissue at the margins of the defect. Consolidation of the islands of tissue appears to continue. The few areas of red are subsurface and represent area of higher translucency, revealing the subchondral tissue.

**Figure 3 ijms-23-02743-f003:**
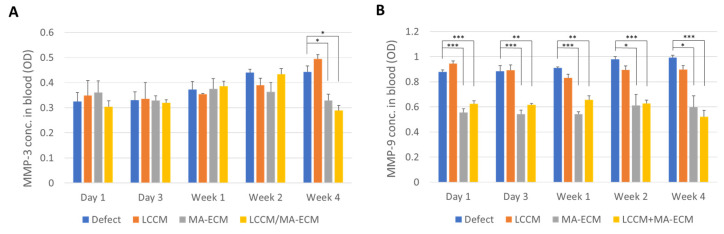
Enzyme-linked immunosorbent assay (ELISA) analysis of MMP-3 (**A**) and -9 (**B**) expressions in serum. The blood samples were centrifuged for serum preparation and cryopreserved at −80 °C until analysis. Serum MMP-3 and -9 were measured by ELISA. Bar graphs describe the expression as mean ± SEM (standard error of the mean); *n* = 2–3 dogs per group. Note that significant difference between MA-ECM containing groups and non-containing groups. Student t-test with Bonferroni correction for multiple comparisons. * *p* < 0.05, ** *p* < 0.01, *** *p* < 0.001.

**Figure 4 ijms-23-02743-f004:**
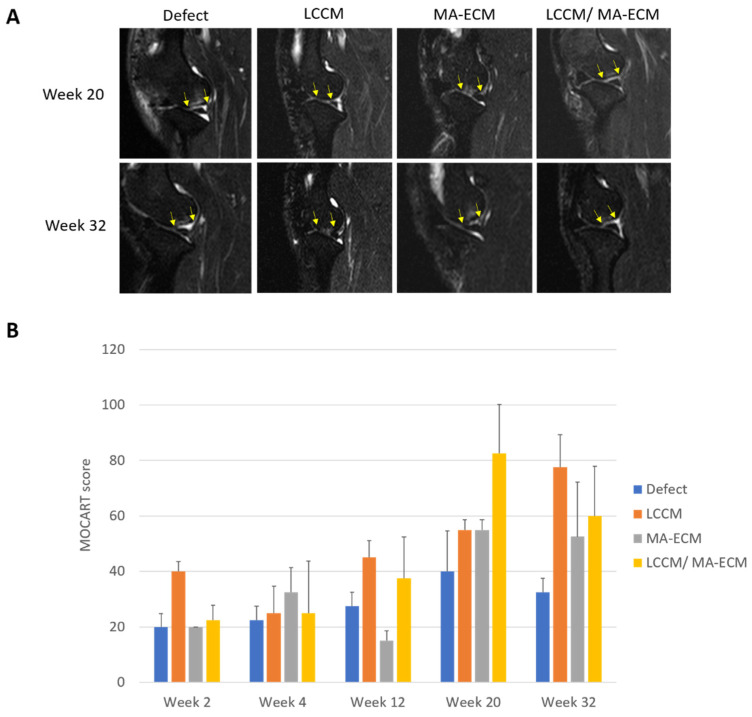
Time-lapse assessment of chondral changes using magnetic resonance imaging (MRI) and MOCART scoring. (**A**) Representative MRI images on weeks 20 and 32. Note that yellow arrows indicate osteochondral alteration among groups. Interestingly, the defect area has been repaired with clear chondral margins from week 20 in LCCM/MA-ECM group. (**B**) The graph shows time-lapse MOCART scores at weeks 2, 4, 12, 20, and 32. *n* = 2–3 dogs per group.

**Figure 5 ijms-23-02743-f005:**
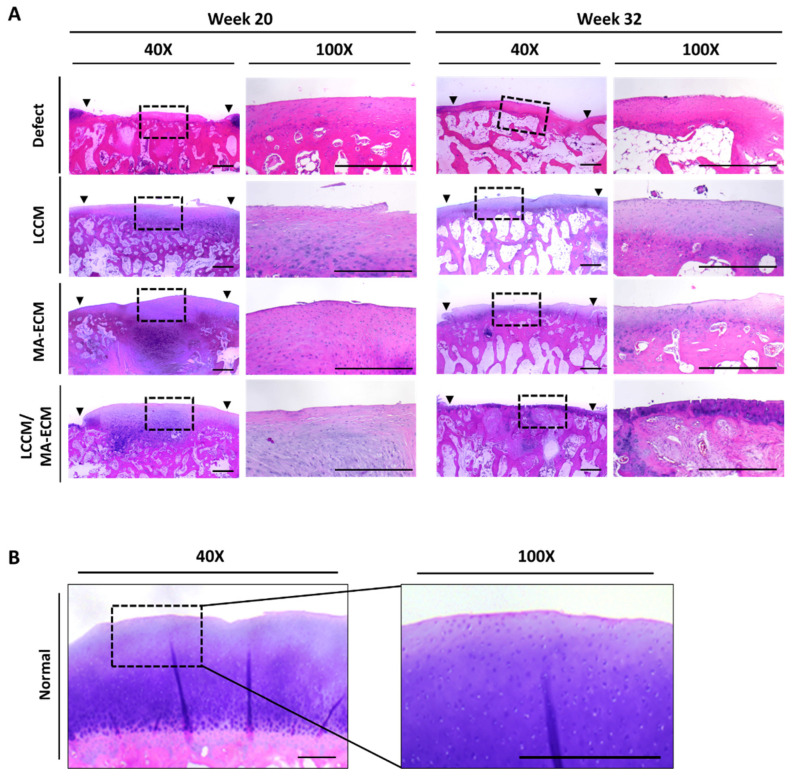
Hematoxylin and eosin (H&E) staining analysis of knee articular cartilages. (**A**) Representative H&E stained images of cartilage on weeks 20 (right knee) and 32 (left knee) in each group. Note that denser cell distributions in MA-ECM and LCCM/MA-ECM groups compared to the other groups. (**B**) H&E stained images of normal cartilage. Black arrows: margins of the osteochondral defect area; black dotted rectangle: low power view (40×) of the area within the rectangle in the right panel (100×). Scale bars = 500 μm.

**Figure 6 ijms-23-02743-f006:**
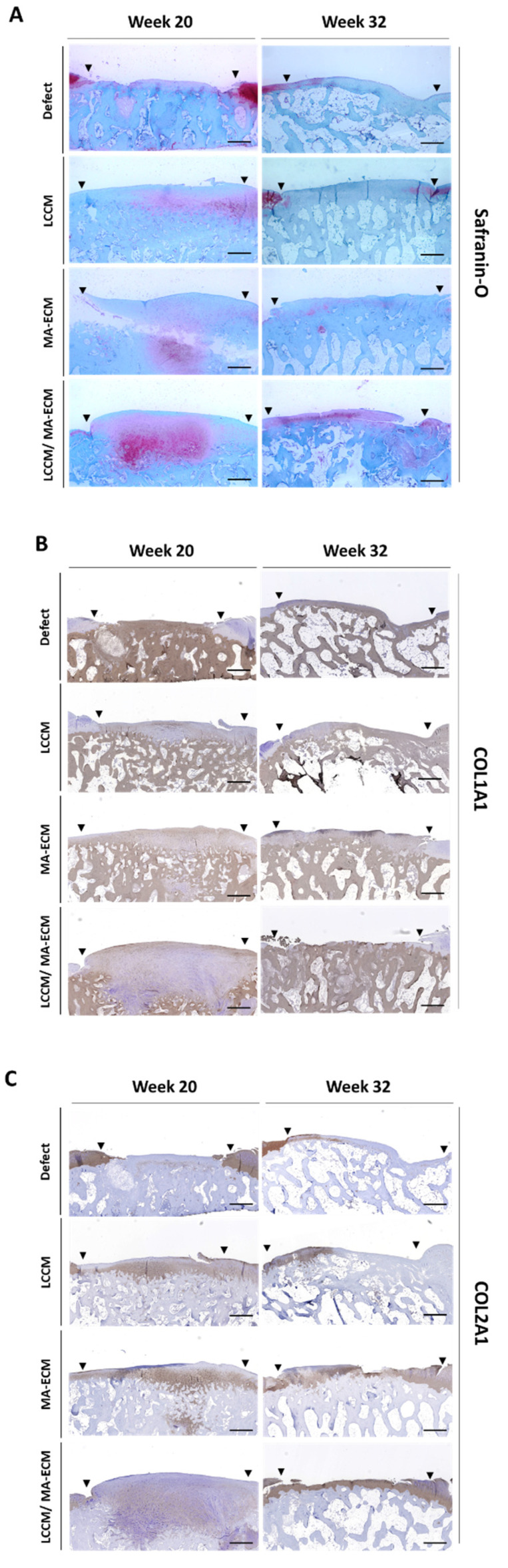
Evaluation of safranin-O (staining) and type 1, 2 collagen (IHC) by light microscopy. 40× representative images result among defect (negative control) and experimental groups of safranin-O (**A**), type 1 collagen—COL1A1 (**B**), type 2 collagen—COL2A1 (**C**) on weeks 20 and 32. Black triangles indicate the margin of chondral defects. Note that intense stain of safranin-O and type 1 collagen in LCCM/MA-ECM group on week 32. Scale bars = 500 um.

**Table 1 ijms-23-02743-t001:** ICRS visual histological scores calculated under microscopy.

Group	Feature(s)	Defect	LCCM	MA-ECM	LCCM/MA-ECM
Week 20	Surface (0–3)	0	3	3	3
	Matrix (0–3)	2	2	1	1
	Cell distribution (0–3)	1	1	2	2
	Cell viability (0–3)	3	1	3	3
	Subchondral bone (0–3)	2	2	2	2
	Cartilage mineralization (0–3)	3	3	3	3
	Total score(s)	11	12	14	14
Week 32	Surface (0–3)	0	3	3	3
	Matrix (0–3)	2	2	2	3
	Cell distribution (0–3)	0	2	2	2
	Cell viability (0–3)	3	3	3	3
	Subchondral bone (0–3)	2	2	2	2
	Cartilage mineralization (0–3)	3	3	3	3
	Total score(s)	10	15	15	16

**Table 2 ijms-23-02743-t002:** The composition of the 3D printed matrix for each group. Defect only group served as a negative control and Exp 1–3 served as the experimental groups. FU = Follow Up.

Group(s)	LCCM	MA-ECM	Study Duration		Number of Animals
Control	−	−	(Right knee) FU week 20	(Left knee) FU week 32	3
Exp 1.	+	−			3
Exp 2.	−	+			3
Exp 3.	+	+			3

## Data Availability

The data presented in this study are available on request from the corresponding author.

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
