# Peer review of "Evaluation of Three-Dimensional Bioprinted Human Cartilage Powder Combined with Micronized Subcutaneous Adipose Tissues for the Repair of Osteochondral Defects in Beagle Dogs"

_ijms, 2022, doi:10.3390/ijms23052743_

Round 1

Reviewer 1 Report

In my opinion, the revised manuscript reaches the standard for publication in IJMS.

Reviewer 2 Report

Tissue engineering is the current research topic in many degenerative disorders, including osteoarthritis. However, how to conduce stem cells/implants to become what we wish is unknown. The study provide some clues about feasibility of 3D printed bio-composite and used several measures to confirm suitability of this model. Very encouraging results for osteoarthritis research.

This manuscript is a resubmission of an earlier submission. The following is a list of the peer review reports and author responses from that submission.

Round 1

Reviewer 1 Report

Dear Authors,

congratulations for your outstanding study. In my opinion it reaches the standard for publication in IJMS in current form. Well done!

BW

Reviewer 2 Report

ijms-1541659-peer-review-v1

Title: Evaluation of Three-dimensional Bio-printed Human Cartilage Powder Combined with Micronized Subcutaneous Adipose Tissues for the Repair of Osteochondral Defects in Beagle Dogs

General comments

This manuscript investigates the ability of 3D printed bio-matrix to repair critical size femoral chondral defects in a canine weight-bearing model. The paper is interesting and well written and covers a key issue in the treatments of articular cartilage defects.

I have only a few minor comments:

  1. Abstract, page 1, lines 26-27: “The experimental……..groups”. The sentence is not clear.
  2. Discussion, page 9, line 260: “dirived”. Please correct.
  3. Histology and immunohistochemistry (IHC) analysis, page 11, line 393: “rinsing” please correct.

Reviewer 3 Report

This manuscript describes a pilot animal study, comparing three different materials developed for regeneration of articular cartilage in dogs. A non-treated, empty defect served as control. The study is of some interest but the design has not been described sufficiently and therefore aim and outcome remains widely unclear. The Results part is completely non-understandable without having read the M&M section before. The Discussions part contains a summary of the own findings but does lack any comparison with results from other labs and therefore does not meet the requirements. The manuscript therefore needs extensive revision.

Detailed comments:

+ Title: “bioprinting”, “bioprinted” etc. is commonly written without hyphen (bio-printed); should be corrected throughout the manuscript; as well: bioink, not bio-ink

+ Please remove „e-mail@e-mail.com“ from affiliation 2 on the 1st page

+ The sentence “The experimental group showed a repair tissue that was much more organized and more similar to normal cartilage compared to the experimental groups” In the abstract makes no sense; probably meant is “…compared to the control groups”-?

+ Please remove “keyword 1” from the keyword list

+ Introduction: A reference for the statement “According to the latest information from the US Center for Disease Control…” needs to be added. What does mean “latest” (which year)-?

+ Page 2, line 60: The statement “For 3D bio printing, biomaterials such as living cell, extracellular matrix (ECM) materials, even soft tissue or bone are used as a bio-ink” is inconsistent. First, cells are no biomaterials and also neither soft tissues nor bone can used as bioinks, only ECM components extracted from those (or cells in combination with materials). This needs revision.

+ Also the following statement is not correct: “However, 3D bio-printing technology has developed improved printing accuracy and precision, enabling more complex tissue or organs to be manufactured”. Bioprinting is significantly less precise compared to common 3D printing technologies like FDM, LSM or SLA as it needs to operate with quite soft bioinks and nozzles which cannot be very small to avoid high shear stress for the cells. In addition, not one single (functional) organ has been bioprinted so far. Such a comparison between material-printing and bioprinting should emphasis that the first is producing implants and the second tissue engineering constructs.

+ Results: The beginning of the results part is completely non-understandable as the authors missed to explain the study design – and the different types of implants that have been investigated in this animal experiment! The results chapter needs to be understandable without having read the M&M part which is definitely not the case here. Please describe your study in detail including explanation of all abbreviations. What does “control” means (Fig. 1): Empty defect or non-operated dogs? All this remains unclear. At which time point the surgery was performed? Day 0, day 1, day 2?

+ Caption of Fig. 1: The statement “The different alphabets indicate significant difference (p<0.05) among experimental groups” remains unclear; alphabetic characters? But why a, bc and c – and why only in the MMP-9 graph? Significant differences between groups should be shown as it is commonly done with * and horizontal brackets.

+ The second paragraph “Mechanical testing” must be 2.2, not 2.1. Again, it is difficult to understand without having read the respective paragraph of the M&M part before.

+ Paragraph 2.3 should be shifted to the beginning of the results part as it would help to understand what has been done – and provides a more fundamental description of the study compared to MMP and mechanical properties of the explants.

+ However, any description of how the different constructs (implants) have been prepared is missing and needs to be added.

+ “Chondral defects were observed at week 20 and 32 after implantation”. As the dogs were euthanized at week 32 (at least this has been stated in the abstract): how the observation at week 20 was done? By arthroscopy? Please describe this properly. Page 3, line 116: with “shinny” probably “shiny” is meant-?

+ Figure 2A: “Defect” = empty control? If I have understood the study design correctly, Fig. 2 shows selected explants of each group, right? Photographs of all explants should be provided in an additional Figure as Supplementary Material.

+ Figure 2B: The statement “The few areas of red are subsurface and are due to the long focal length of the dissecting microscope” does not become clear to me. These structures look like blood capillaries-? Please explain better what can be seen here; at least, this are not optical artefacts.

+ 2.4 MRI. Please add a reference for the MOCART scoring system. Fig. 3: Please add the information about the number of animals per group that has been investigated by MRI. Are there any significant differences between the scores of the 4 groups at any of the time points?

+ 2.5 Histology. Please add a reference for the ICRS visual histological assessment scale. In addition, a bit more explanation on that is needed: Zero points is the lowest possible score – but what is the highest score (i.e. that for healthy, native cartilage)? Otherwise the results shown in Table 1 are not very meaningful.

+ For better comparison, histological images of healthy knee cartilage of beagles should be included in Fig. 4. If possible, the thickness of the newly formed cartilage (in comparison to the natural thickness of the healthy tissue) should be quantified. Am I right that the thickness of the newly formed cartilage layer of all three experimental groups is diminishing from week 20 to week 32 – in contrast to the empty defects where the thickness is increasing? Could you please comment on that-?

+ 2.5 Safranin-O staining & immunohist. The micrographs of Fig. 5 are too small and hard to see properly. This figure therefore should be reformatted (e.g. by positioning of the micrograph panels a, b and c vertically and not horizontally on the page). As already mentioned above (HE staining), stainings of healthy cartilage should be included in Fig. 5 for better comparison. What is meant by “> 75%” etc.-? 75% of the animals of one group – or 75% of the surface area of the defect dimensions, visible in the histological sections? This remains unclear. If the surface area is meant: how the 100% has been defined? This might be very complicated as the surface of the regenerated cartilage is quite uneven in most animals.

+ I know Safranin-O as staining for glycosaminoglycans (GAG) in articular cartilage histology; why you are speaking about “peptidoglycans”-?

+ Table 2: Please explain the scoring system you have used here. If not defined by yourself a reference needs to be given.

+ Discussion: Page 8, line 245 “mechanical texting” should be “testing”

+ The discussion part has been written like a summary of the own results but does lack any comparison with the multiple already published studies of other researchers on 3D bioprinted articular cartilage constructs, studies about utilization of MSC in comparison to chondrocytes and also animal experiments (including large animals like horses, e.g. performed by Jos Malda and his group).

+ Materials and Methods: Here for the first time the composition of the implanted materials is (briefly) described. This forces the reader to jump to the M&M part before one is able to understand the Results part – which is not useful. As mentioned earlier, the study design and material composition has to be explained properly at the beginning of the Results part. I want to recommend to shift Figure 6 to the beginning of the Results section (as new Figure 1) as it helps to understand the design of the animal trial.

+ Animal study: as you had 3 animals per group and 2 time points: how many dogs per group were sacrificed at 20 and 32 weeks, respectively? Please add this information.

+ In Table 3 three conditions are described as controls and only one as “experimental group” (but with No. 3-?). For my understanding, only the empty defect should be classified as “control” and all others as “experimental groups 1-3”. This definition should be used throughout the whole manuscript.

+ 4.4 Bioink preparation: Here, MA-ECM is defined as “Minimally manipulated Autologous- Extracellular Matrix”c – whereas in the Introduction of the manuscript the same abbreviation is described as “cells from human adipose tissue”-? Even after having read paragraph 4.4.1 it remains unclear to me whether MA-ECM contains any human cells – or only ECM components. As ECM is defined as the cell-free component of a tissue this needs to be clarified carefully. However, if (live?) human cells (MSC or adipocytes – or both?) are part of the “MA-ECM”: why these cells did not lead to an immunological rejection after implantation into dogs?

+ However, if none of the “bioinks” contained live cells neither the term bioink nor bioprinting may not be used as by definition a bioink needs to contain living cells. Otherwise, it must be called “biomaterial ink” (and the process 3D-printing and not bioprinting). Please refer to J. Groll et al.: A Definition of Bioinks and their Distinction from Biomaterial Inks. Biofabrication 2019, 11, 013001

+ For me it remains unclear why 3D printing has been used to prepare the gels for implantation. Obviously, viscosity was not suitable for printing and therefore these PCL molds had to be printed in advance. The implants therefore could have been prepared also by pipetting into these molds which would have led to the same results. Could you please comment on that-?

+ 4.5.2: What is meant by “The dual printing of Bio-ink 1 and 2…”? Were both “inks” mixed during the extrusion (printing) process? How was that achieved? By using a coaxial nozzle – or a static mixer? Or has the printer two printing channels and the printing of the two components (inks) was done sequentially? By the way, the thrombin solution should not be called “ink” as it is acting as crosslinker only for the fibrinogen-based materials = “Bioink 1”…

+ Page 13, line 447 “inturpretation” must be “interpretation”